# Prevalence of heated tobacco product use among adolescents in Taiwan

**Li-Chuan Chang**[1]☯**, Yue-Chune Lee**[1,2]☯**, Chieh Hsu**[1]**, Pei-Ching Chen**[iD][3]*

**1** Institute of Health and Welfare Policy, School of Medicine, National Yang-Ming University, Taipei, Taiwan, **2** Master Program on Trans-Disciplinary Long-Term Care and Management, National Yang-Ming University, Taipei, Taiwan, **3** Department of Health and Welfare, College of City Management, University of Taipei, Taipei, Taiwan

☯ These authors contributed equally to this work.
* peiching@ntu.edu.tw

**Data Availability Statement:** The Global Youth Tobacco Survey (GYTS) is conducted and all data is owned by the Health Promotion Administration, Ministry of Health and Welfare Taiwan. All personally identifiable information was removed

## Abstract

### Objective

To determine the penetration of heated tobacco products (HTPs) into the youth market in Taiwan, with a particular focus on the correlation between IQOS use and the usage of other tobacco products.

### Methods

Data from the 2018 Global Youth Tobacco Survey were used to assess previous experience with and current use (within 30 days prior to survey completion) of IQOS products by Taiwanese students aged 12–18 years. Independent variables included the usage patterns of conventional cigarettes and e-cigarettes. The control variables included background information (gender, grade, monthly income/allowance, household educational level, smoking status at home and among close friends), access to free cigarettes, as well as exposure to cigarette advertisements and anti-tobacco courses. Logistic regression was used to identify tobacco usage patterns correlated with IQOS use.

### Results

In 2018, 2.33% of Taiwan's adolescents were currently using IQOS and 4.17% had tried IQOS. The use of conventional cigarettes and e-cigarettes (individually and together) were associated with an elevated risk of the ever use and current use of IQOS.

### Conclusion

Despite the fact that HTP products are not sold legally in Taiwan, the use of IQOS products by young people is far from negligible. We recommend amending the "Tobacco Hazards Prevention Act" to include regulations pertaining to the sale and marketing of HTPs.

from the data before being made available to the authors. Furthermore, all of the research participants signed contracts ensuring the security of the data. Thus, legal restrictions prevent the authors from sharing the raw data. Nonetheless, any interested researcher may apply to the Health Promotion Administration for access to GYTS data. Contact Address of the Health Promotion Administration, Ministry of Health and Welfare of Taiwan: No.36, Tacheng St., Datong Dist., Taipei City 10341, Taiwan (R.O.C.); 886-2-25220888 Application URL: https://www.hpa.gov.tw/Pages/List.aspx?nodeid=1726.

**Funding:** This study was funded by the Health and Welfare Surcharge of Tobacco Products, the Health Promotion Administration, Ministry of Health and Welfare, Taiwan (No. G1071219, No. G1071219-109). The Health Promotion Administration, Ministry of Health and Welfare conducted the Taiwan Global Youth Tobacco Survey, and provided the data for use in this study.

**Competing interests:** The authors have declared that no competing interests exist.

## Introduction

Heated Tobacco Products (HTPs) heat tobacco to a lower temperature than that of a combusted cigarette to create an aerosol that the user inhales [1]. Unlike e-cigarettes, which heat a nicotine-containing liquid solution, HTPs heat tobacco sticks, capsules, or pods containing processed tobacco [2]. In November 2014, the tobacco giant Philip Morris International introduced in Japan a new type of HTP named IQOS [3].

Students and young adults are prime targets in the design and marketing of HTPs [4], and it appears that those efforts are paying off [5, 6]. In 2017, Kim et al. conducted an online survey of young adults in South Korea aged 19–24 years. They found that nearly 40% of the respondents had heard of IQOS, roughly 6% had tried the product, and 3.5% were current users [7]. Based on data obtained in England, Canada, and the USA in 2017, Czoli et al. [8]. reported that roughly 7.0% of youths (England = 5.6%, Canada = 6.4%, and USA = 9.1%) aged 16–19 years were aware of IQOS products and 38.6% of those (England = 41.8%, Canada = 33.0%, and USA = 40.9%) were interested in trying the product. Dai [9] reported that in 2019, self-reported awareness, ever use, and current use of HTPs among US students were 12.8%, 2.4%, and 1.6%, respectively. In 2017, Liu et al. [10] reported that among young adults in Italy (aged ≥15 years), 19.5% were aware of IQOS, 1.4% had tried it, and 2.3% were intending to try it.

Despite the fact that HTP products are not sold legally in Taiwan, many young people still gain access to these items through online sales channels, such as IQOS websites, Facebook or LINE groups. HTP products are easily purchased online, and a number of large-scale e-cigarette or IQOS stores are currently operating brick-and-mortar stores in Taiwan. IQOS is legally sold in Asian countries such as Japan and South Korea [2], and at present, those countries are the primary source of black market IQOS in Taiwan. Most of the online sales of IQOS sold targeting Taiwanese customers are based in Japan. The situation of young people using IQOS is clearly an issue worthy of concern. In this study, data from the Taiwan Global Youth Tobacco Survey were used to determine the penetration of heated tobacco products (HTPs) into the youth market in Taiwan, while identifying factors correlated with previous experience and current usage of IQOS by high school students.

## Materials and methods

### Data sources

This study was based on data from the 2018 Taiwan Global Youth Tobacco Survey (GYTS) [11], which is conducted annually in junior and senior high schools throughout the country by the Health Promotion Administration of Taiwan. The primary objective of the survey is to monitor tobacco use among students aged 12–18 years, while permitting statistical comparison with data obtained in other countries [12]. Questions pertaining to e-cigarette and IQOS use have been included since 2014 and 2018, respectively.

Local health bureaus administer the survey in classrooms in the form of a paper survey after undergoing seminars on the presentation of questionnaires by the Health Promotion Administration [13]. The parents of students are informed in advance that they have the right to withhold consent to participate in the survey. All eligible students in the school take the survey at the same time to prevent discussion. The questionnaire is self-administered and anonymous. The GYTS is an official investigation conducted annually by the Health Promotion Administration, having passed an IRB review (EC1041011-F-R2).

All of the data used in this study were obtained by the Health Promotion Administration, Ministry of Health and Welfare, Taiwan. None of the authors were involved in establishing testing protocols, drawing up research questions, or recruiting students. Data made available

to the authors had all of the links to personal information removed to ensure the anonymity of students. Ethical approval for the current study was granted by the Institutional Review Board of National Yang-Ming University (YM108025E).

## Study population

Protocols established by the Health Promotion Administration stipulate that sampling begins with the selection of schools, followed by classes in those schools, and finally students. Each local government (at the city or county level) is tasked with recruiting 900 students at the junior high level (grades 7–9) and 1,100 students from senior high (grades 10–12). In 2018, a total of 49,971 sample students participated in the survey: junior high (22,693) and senior high (27,278). This yielded 44,905 valid surveys: junior high (20,966) and senior high (23,939). The overall completion rate was 89.86%: junior high (92.39%) and senior high (87.76%). Samples that were missing answers related to the previous or current use of IQOS were excluded. A total of 44,757 students answered questions related to previous experience with IQOS and 44,847 answered questions related to current IQOS use.

Sample weighting was used to adjust for missing responses (at the school, class, and student levels) and sample selection (at the school and class levels). The sample population also underwent post-stratification to adjust for grade and gender distributions in the total population. After sample weighting, the number of students who responded to the ever use IQOS item ("Have you ever used an IQOS?") was equivalent to 1,498,502 in the general population. Likewise, the number of students who responded to the current use IQOS item ("During the past 30 days, have you used an IQOS?") was equivalent to 1,501,527 students in the general population. Fig 1 illustrates the sampling selection process.

## Variables

The dependent variables of IQOS usage were estimated using the following items: (1) "Have you previously tried an IQOS product? (yes/no)" and (2) "Have you used an IQOS product in the last 30 days? (yes/no)."

Two logistic regression models were employed in this study. The first model dealt separately with the independent variables (i.e., the use of conventional cigarettes OR the use of e-cigarettes). Instances of cigarette usage were identified using the following item: "Have you used a cigarette in the last 30 days? (yes/no)." Instances of e-cigarette were identified using the following item: "Have you used an e-cigarette in the last 30 days? (yes/no)." Note however that this approach did not provide a comprehensive indication of smoking behavior, as many of the participants were likely to use both.

We therefore established a second model in which the independent variables were combined as a single variable indicating smoking in general, which was then divided into the following four categories: cigarette use only, e-cigarette use only, dual-use, and none.

The control variables included background information (gender, grade, monthly income/allowance, household educational level, household smoking status, and close friends smoking status), access to free cigarettes, as well as exposure to cigarette advertisements and anti-tobacco courses.

Monthly allowance was categorized as follows: USD 0, USD≦50, USD 50–100, and USD≧100. Household education level was based on the parent who achieved the highest from the following: completed junior high school, completed senior high school, completed college, and post-graduate. Household smoking status was categorized as follows: only the father smokes, only the mother smokes, both parents smoke, and neither parent smokes. Smoking

**Rules for Sampling Adolescent population**

1. 900 students from grades 7-9 in each county/city

2. 1,100 students from grades 10-12 in each county/city

**Initial sample size: 49,971**

1. Grades 7-9: 22,693

2. Grades 10-12: 27,278

**Valid completed samples: 44,905**
**Completion rate: 89.86%**

3. Grades 7-9: 20,966; 92.39%

4. Grade 10-12: 23,939; 87.76%

**Exclude those with missing answers on the previous or current use of IQOS**

**Study sample**

5. Previous use of IQOS: 44,757

6. Current use of IQOS: 44,847

**Final weighted sample**

7. Previous use of IQOS: 1,498,502

8. Current use of IQOS: 1,501,527

**Fig 1. Flow chart illustrating sampling process.**

status among close friends was categorized as having at least one close friend who smokes or having no close friends who smoke.

Access to free tobacco products was determined using the following item: "Has any tobacco company ever offered you a free tobacco product? (yes/no)." Exposure to cigarette advertisements was based on whether students had seen cigarette advertisements in stores. Access to anti-tobacco courses was determined using the following item: "Have you received instruction as to the dangers of tobacco use within the last year? (yes/no)."

### Analysis

All statistical analysis was conducted using SAS software version 9.4 (SAS Institute Inc., Cary, NC, USA), where a $P<0.05$ indicated results of statistical significance. Following adjustment in accordance with the sample design (i.e., county/city and school), the Rao-Scott chi-square test was used to compare the demographic characteristics of the following groups of students: (1) Previously tried IQOS vs. Never previously tried IQOS and (2) Currently using IQOS vs. Not currently using IQOS. Logistic regression was also adjusted in accordance with the sample design (i.e., county/city and school) to identify independent variables with a significant correlation to the previous or current use of IQOS products.

### Results

Table 1 lists the demographic characteristics of the study participants in the following three categories: students who had previously tried IQOS, students who had never tried IQOS, and overall. The percentage of students who had previously tried IQOS was 4.17%. The students who were most likely to have previously tried IQOS were those who had used both cigarettes AND e-cigarettes (i.e., dual-use) in the previous 30 days, followed respectively by e-cigarette use only, cigarette use only, and those who had used neither.

Note that the percentage of students who had previously tried IQOS was higher among boys, students in grade 12, those with a high monthly allowance (USD≧116.04), and those whose parents had a low educational level (junior high school and below). The likelihood of trying IQOS was higher among students living in households where both parents were smokers and/or their close friends were smokers. The likelihood of trying IQOS was also higher among students with access to free cigarettes, those who had been exposed to cigarette advertisements, and those who had attended anti-tobacco classes.

Table 2 lists the demographic characteristics of the study participants in the following three categories: students who were currently using IQOS, students who were not currently using IQOS, and overall. The percentage of students who were currently using IQOS was 2.33%. The overall results for the current use of IQOS were essentially the same as those for ever use. The students who were most likely to be currently using IQOS were those who had used both cigarettes AND e-cigarettes (i.e., dual-use) in the previous 30 days, followed respectively by e-cigarette use only, cigarette use only, and those who had used neither.

Table 3 lists the results of the two logistic regression models illustrating the factors correlating with the ever use of IQOS. Based on model 1, it was found that students who had used cigarettes or e-cigarettes in the previous 30 days were more likely than their non-smoking counterparts to have previously tried IQOS. Furthermore, previous experience with IQOS was more strongly correlated with cigarette use than with e-cigarette use. Based on model 2, it was found that students who had used both cigarettes and e-cigarettes in the past 30 days were the most likely to have previously tried IQOS, followed respectively by e-cigarette use only, cigarette use only, and none.

**Table 1. Demographic characteristics of students who had previously tried IQOS, those who had never tried IQOS, and overall.**

| Characteristic | Overall | | Previously tried IQOS | | p |
|---|---|---|---|---|---|
| | N [a] | (%) [b] | Yes (%) [c] | No (%) [c] | |
| Total | 1498502 | | 4.17 [c] | 95.83 [c] | |
| Gender | | | | | ‡ |
| Boy | 777923 | 51.99 | 5.02 | 94.98 | |
| Girl | 718363 | 48.01 | 3.24 | 96.76 | |
| Grade | | | | | ‡ |
| 7 | 214508 | 14.31 | 3.74 | 96.26 | |
| 8 | 229287 | 15.30 | 3.46 | 96.54 | |
| 9 | 241815 | 16.14 | 4.08 | 95.92 | |
| 10 | 280968 | 18.75 | 4.33 | 95.67 | |
| 11 | 276570 | 18.46 | 3.74 | 96.26 | |
| 12 | 255355 | 17.04 | 5.56 | 94.44 | |
| Monthly Allowance | | | | | ‡ |
| USD 0 | 169608 | 11.37 | 4.54 | 95.46 | |
| USD ≦50 | 744025 | 49.86 | 3.25 | 96.75 | |
| USD 50–100 | 335345 | 22.47 | 3.72 | 96.28 | |
| USD ≧100 | 243272 | 16.30 | 7.31 | 92.69 | |
| Use of cigarettes in the previous month | | | | | ‡ |
| Yes | 80984 | 5.50 | 16.02 | 83.98 | |
| No | 1392770 | 94.50 | 3.29 | 96.71 | |
| Use of e-cigarettes in the previous month | | | | | ‡ |
| Yes | 38038 | 2.57 | 18.79 | 81.21 | |
| No | 1441918 | 97.43 | 3.68 | 96.32 | |
| Combined use of cigarettes and e-cigarettes in the previous month | | | | | ‡ |
| Use of both cigarettes and e-cigarettes | 19000 | 1.30 | 21.47 | 78.53 | |
| Use of e-cigarettes only | 16223 | 1.11 | 13.4 | 86.6 | |
| Use of cigarettes only | 59111 | 4.06 | 14.41 | 85.59 | |
| None | 1362244 | 93.52 | 3.1 | 96.9 | |
| Educational level of parents | | | | | † |
| Junior high school and below | 123057 | 8.91 | 5.65 | 94.35 | |
| Senior high school | 542652 | 39.29 | 4.28 | 95.72 | |
| University or college | 540821 | 39.16 | 3.64 | 96.36 | |
| Graduate school | 174624 | 12.64 | 3.98 | 96.02 | |
| Smoking status at home | | | | | ‡ |
| Both parents smoke | 121300 | 8.27 | 6.28 | 93.72 | |
| Only the father smokes | 480330 | 32.73 | 3.72 | 96.28 | |
| Only the mother smokes | 29534 | 2.01 | 4.65 | 95.35 | |
| Neither smokes | 836346 | 56.99 | 3.99 | 96.01 | |
| Smoking status of close friends | | | | | ‡ |
| Yes | 631929 | 42.27 | 5.25 | 94.75 | |
| No | 863098 | 57.73 | 3.30 | 96.70 | |
| Access to free cigarettes | | | | | ‡ |
| Yes | 45278 | 3.06 | 21.03 | 78.97 | |
| No | 1435534 | 96.94 | 3.54 | 96.46 | |
| Exposure to cigarette advertisements | | | | | † |
| Yes | 342036 | 22.89 | 4.91 | 95.09 | |
| No | 1152176 | 77.11 | 3.88 | 96.12 | |

(*Continued*)

**Table 1.** (Continued）

| Characteristic | Overall | | Previously tried IQOS | | p |
|---|---|---|---|---|---|
| | N [a] | (%) [b] | Yes (%) [c] | No (%) [c] | |
| Exposure to anti-tobacco courses | | | | | * |
| Yes | 707083 | 47.27 | 4.47 | 95.53 | |
| No | 788642 | 52.73 | 3.83 | 96.17 | |

[a] The total numbers were not equal in each variable due to missing value(s).

[b] Summed percentages in each column.

[c] Summed percentages in each row.

P: The Rao-Scott chi-square test was used to test the difference between independent variables and previous use of IQOS

‡ p-values <0.001

†p-values <0.01

*p-values <0.05.

Overall, the results of the control variables were the same under model 1 and model 2. The following factors were positively correlated with the likelihood of previously trying IQOS: male gender, having close friends who smoke, obtaining free cigarettes, and having learned about the dangers of tobacco. We observed a nonlinear relationship between monthly allowance and previous use of IQOS. Students with low (USD≦50) or moderate (USD 50–100) monthly allowance were less likely to have tried IQOS than were students with high (USD≧100) monthly allowance. We observed no difference between students with no monthly allowance (USD 0) and those with a high (USD≧100) monthly allowance in terms of previously trying IQOS. Table 4 lists two logistic regression models illustrating the factors correlating with the current use of IQOS. Based on model 1, it was found that students who had used cigarettes or e-cigarettes in the past 30 days were more likely than their non-smoking counterparts to be currently using IQOS. Furthermore, current IQOS use was more strongly correlated with e-cigarette use than with cigarette use. Based on model 2, it was found that students who had used both cigarettes and e-cigarettes in the past 30 days were the most likely to be currently using IQOS, followed by e-cigarette use only, cigarette use only, and none.

The results of control variables were the same under model 1 and model 2. The following factors were positively correlated with the likelihood of current IQOS use: male gender, having close friends who smoke, and access to free cigarettes. The following factors were negatively correlated with the likelihood of current IQOS use: higher grades (grade 10–12) and low (USD≦50) monthly allowance.

## Discussion

Our results revealed that in 2018, roughly 4% of high school students in Taiwan had previously tried IQOS products and 2% were currently using them. The current use of cigarettes and the current use of e-cigarettes were the factors most strongly correlated with the likelihood of previously trying IQOS and the likelihood of currently using IQOS. The combined use of cigarettes and e-cigarettes presented the strongest correlation, followed respectively by the use of e-cigarettes only, and the use of cigarettes only. Adolescents who did not use any tobacco products were very unlikely to try IQOS.

The usage statistics in the current study are slightly lower than those in a report on young Korean adults by Kim et al., in which 5.7% of the study population had tried IQOS and 3.5% were currently using IQOS [7]. The differences can be attributed largely to the study methods (school-based survey vs. online survey), the size of the sample, and the age of the respondents

**Table 2. Demographic characteristics distribution of students who were currently using IQOS, those who were not using IQOS, and overall.**

| Characteristic | All | | Current use of IQOS | | p |
|---|---|---|---|---|---|
| | N [a] | (%) [b] | Yes (%) [c] | No (%) [c] | |
| Total | 1501527 | | 2.33 [c] | 97.67 [c] | |
| Gender | | | | | ‡ |
| Boy | 779780 | 52.03 | 3.01 | 96.99 | |
| Girl | 718800 | 47.97 | 1.47 | 98.53 | |
| Grade | | | | | * |
| 7 | 214748 | 14.30 | 1.99 | 98.01 | |
| 8 | 229647 | 15.29 | 1.91 | 98.09 | |
| 9 | 241913 | 16.11 | 1.96 | 98.04 | |
| 10 | 281736 | 18.76 | 2.69 | 97.31 | |
| 11 | 277029 | 18.45 | 2.27 | 97.73 | |
| 12 | 256453 | 17.08 | 3.01 | 96.99 | |
| Monthly Allowance | | | | | ‡ |
| USD 0 | 170113 | 11.38 | 2.14 | 97.86 | |
| USD ≦50 | 744969 | 49.82 | 1.48 | 98.52 | |
| USD 50–100 | 335925 | 22.46 | 2.30 | 97.70 | |
| USD ≧100 | 244405 | 16.34 | 5.05 | 94.95 | |
| Use of cigarettes in the previous month | | | | | ‡ |
| Yes | 83056 | 5.63 | 15.58 | 84.42 | |
| No | 1393088 | 94.37 | 1.28 | 98.72 | |
| Use of e-cigarettes in the previous month | | | | | ‡ |
| Yes | 40470 | 2.73 | 30.70 | 69.30 | |
| No | 1442738 | 97.27 | 1.35 | 98.65 | |
| Combined use of cigarettes and e-cigarettes in the previous month | | | | | ‡ |
| Use of both cigarettes and e-cigarettes | 20419 | 1.40 | 33.76 | 66.24 | |
| Use of e-cigarettes only | 16541 | 1.13 | 18.90 | 81.10 | |
| Use of cigarettes only | 59591 | 4.08 | 8.44 | 91.56 | |
| None | 1362451 | 93.38 | 0.97 | 99.03 | |
| Educational level of parents | | | | | ‡ |
| Junior high school and below | 123637 | 8.93 | 3.72 | 96.28 | |
| Senior high school | 543645 | 39.28 | 2.69 | 97.31 | |
| University or college | 541726 | 39.14 | 1.68 | 98.32 | |
| Graduate school | 175088 | 12.65 | 2.07 | 97.93 | |
| Smoking status at home | | | | | ‡ |
| Both parents smoke | 121901 | 8.29 | 4.73 | 95.27 | |
| Only the father smokes | 480892 | 32.72 | 2.27 | 97.73 | |
| Only the mother smokes | 29853 | 2.03 | 5.53 | 94.47 | |
| Neither smokes | 837168 | 56.96 | 1.70 | 98.30 | |
| Smoking status of close friends | | | | | ‡ |
| Yes | 633620 | 42.32 | 3.72 | 96.28 | |
| No | 863766 | 57.68 | 1.15 | 98.85 | |
| Access to free cigarettes | | | | | ‡ |
| Yes | 46068 | 3.11 | 11.14 | 88.86 | |
| No | 1435773 | 96.89 | 1.78 | 98.22 | |
| Exposure to cigarette advertisements | | | | | ‡ |
| Yes | 342874 | 22.92 | 2.93 | 97.07 | |
| No | 1153174 | 77.08 | 1.94 | 98.06 | |

*(Continued)*

**Table 2.** (Continued)

| Characteristic | All | | Current use of IQOS | | p |
|---|---|---|---|---|---|
| | N [a] | (%) [b] | Yes (%) [c] | No (%) [c] | |
| Exposure to anti-tobacco courses | | | | | |
| Yes | 707812 | 47.26 | 2.09 | 97.91 | |
| No | 789882 | 52.74 | 2.30 | 97.70 | |

[a] The total numbers were not equal in each variable due to missing value(s).

[b] Summed percentages in each column.

[c] Summed percentages in each row.

P: The Rao-Scott chi-square test was used to test the difference between independent variables and current use of IQOS

‡ p-values <0.001

†p-values <0.01

*p-values <0.05.

(12–18 years vs. 19–24 years). Nonetheless, we believe that the main reason for the discrepancy is a difference in attitudes toward IQOS use. IQOS products were introduced in Korea in 2017, and the devices are currently available in convenience stores. By contrast, the sale of HTPs is not legally sanctioned in Taiwan, with the result that people must resort to online sales or illegal purchases. It is possible that the price difference may account for some of the differences in adoption. South Korea has legalized the use of IQOS; therefore, the price is lower. These products are still illegal in Taiwan; therefore, the price is higher.

The usage statistics in the current study are slightly higher than those in Dai's 2019 report [9] on US students, in which 2.4% of the study population had tried HTPs (including iQOS, glo, and Eclipse) and 1.6% were currently using HTPs. Dai [9] reported that IQOS was only authorized for sale in the US during the survey period (February 15, 2019 –May 24, 2019) and that IQOS was not launched in the US until October 2019. At the time of that study, the sale of IQOS was not legally sanctioned in Taiwan or the US; however, usage rates were higher in Taiwan than in the US. Clearly, the use of IQOS products by young people in Taiwan is far from negligible.

Not surprisingly, it was observed that current smokers (cigarettes and/or e-cigarettes) were more likely than their non-smoking counterparts to have tried IQOS or currently use IQOS products. These results are similar to those obtained by Dai [9] in which the current users of cigarettes, e-cigarettes, and other tobacco products were more likely than non-users to have tried or be currently using HTPs. Kim et al. [7] reported a similar connection. It is also unsurprising that students who were using both cigarettes and e-cigarettes were more likely to have previously tried IQOS or currently be using IQOS. Nonetheless, we were surprised to note that that those who were using e-cigarettes only were more likely than those using cigarettes only to try IQOS. It appears that one's proclivity to using e-cigarettes may carry over to IQOS products as well. We therefore recommend that governments closely monitor e-cigarettes use.

Male students were more likely than their female counterparts to try IQOS and more likely to be currently using IQOS. Kim et al. [7], Czoli et al. [8], and Dai [9] reported the same link. Note that the link between low/moderate monthly allowance and a lower likelihood of trying IQOS was observed only in the current study.

It appears that IQOS use can largely be explained by peer effects (having smokers as close friends) and environmental factors (access to free cigarettes). In fact, these two effects are probably linked. Note that the survey specified free cigarettes from tobacco companies; however, it is very likely that students obtain free tobacco products from their friends, particularly

**Table 3. Logistic regression results for prior use of IQOS among Taiwanese youth (2018).**

| Independent Variable | Y = Previously tried IQOS | | | | | |
|---|---|---|---|---|---|---|
| | Model 1 | | | Model 2 | | |
| | aOR | 95%CI | | P | aOR | 95%CI | | P |
| Use of cigarettes in the previous month | | | | | | | | |
| Yes | 3.40 | 2.61 | 4.43 | ‡ | | | | |
| No | 1 | | | | | | | |
| Use of e-cigarettes in the previous month | | | | | | | | |
| Yes | 1.98 | 1.38 | 2.84 | ‡ | | | | |
| No | 1 | | | | | | | |
| Combined use of cigarettes and e-cigarettes in the previous month | | | | | | | | |
| Use of both cigarettes and e-cigarettes | | | | | 4.79 | 3.20 | 7.18 | ‡ |
| Use of e-cigarettes only | | | | | 4.11 | 2.60 | 6.50 | ‡ |
| Use of cigarettes only | | | | | 4.10 | 3.16 | 5.32 | ‡ |
| None | | | | | 1 | | | |
| Gender | | | | | | | | |
| Boy | 1.29 | 1.11 | 1.51 | † | 1.28 | 1.10 | 1.49 | † |
| Girl | 1 | | | | 1 | | | |
| Grade | | | | | | | | |
| 7 | 1 | | | | 1 | | | |
| 8 | 0.88 | 0.64 | 1.23 | | 0.88 | 0.63 | 1.22 | |
| 9 | 1.07 | 0.74 | 1.55 | | 1.07 | 0.74 | 1.54 | |
| 10 | 0.94 | 0.67 | 1.32 | | 0.93 | 0.66 | 1.31 | |
| 11 | 0.72 | 0.51 | 1.04 | | 0.72 | 0.51 | 1.04 | |
| 12 | 1.13 | 0.74 | 1.74 | | 1.13 | 0.73 | 1.73 | |
| Monthly Allowance | | | | | | | | |
| USD 0 | 1.03 | 0.76 | 1.40 | | 1.05 | 0.77 | 1.43 | |
| USD ≦50 | 0.73 | 0.59 | 0.91 | † | 0.74 | 0.60 | 0.92 | † |
| USD 50–100 | 0.79 | 0.64 | 0.97 | * | 0.80 | 0.65 | 0.98 | * |
| USD ≧100 | 1 | | | | 1 | | | |
| Educational level of parents | | | | | | | | |
| Junior high school and below | 1 | | | | 1 | | | |
| Senior high school | 0.87 | 0.67 | 1.13 | | 0.87 | 0.67 | 1.12 | |
| University or college | 0.82 | 0.63 | 1.08 | | 0.82 | 0.63 | 1.08 | |
| Graduate school | 0.84 | 0.61 | 1.17 | | 0.84 | 0.61 | 1.17 | |
| Smoking status at home | | | | | | | | |
| Both parents smoke | 1.03 | 0.81 | 1.32 | | 1.03 | 0.80 | 1.31 | |
| Only the father smokes | 0.92 | 0.75 | 1.11 | | 0.91 | 0.75 | 1.10 | |
| Only the mother smokes | 0.77 | 0.50 | 1.19 | | 0.75 | 0.49 | 1.16 | |
| Neither smokes | 1 | | | | 1 | | | |
| Smoking status of close friends | | | | | | | | |
| Yes | 1.20 | 1.01 | 1.41 | * | 1.17 | 0.99 | 1.39 | |
| No | 1 | | | | 1 | | | |
| Access to free cigarettes | | | | | | | | |
| Yes | 4.49 | 3.34 | 6.03 | ‡ | 4.52 | 3.37 | 6.08 | ‡ |
| No | 1 | | | | 1 | | | |
| Exposure to cigarette advertisements | | | | | | | | |
| Yes | 1.10 | 0.94 | 1.30 | | 1.09 | 0.93 | 1.28 | |
| No | 1 | | | | 1 | | | |

*(Continued)*

**Table 3.** (Continued)

| Independent Variable | Y = Previously tried IQOS | | | | | | |
|---|---|---|---|---|---|---|---|
| | Model 1 | | | | Model 2 | | |
| | aOR | 95%CI | | P | aOR | 95%CI | | P |
| Exposure to anti-tobacco courses | | | | | | | |
| Yes | 1.33 | 1.14 | 1.55 | ‡ | 1.33 | 1.14 | 1.55 | ‡ |
| No | 1 | | | | 1 | | | |

aOR: adjusted odds ratios.

Model 1: Determining the use of cigarettes OR e-cigarettes.

Model 2: Considering the use of cigarettes AND e-cigarettes.

*$P<0.05$

†$P<0.01$

‡$P<0.001$.

the first time that they try them. Contrary to expectations, we did not observe a significant correlation between IQOS use and attending classes on the dangers of tobacco. This may be due to the fact that most anti-smoking courses focus on the hazards of conventional tobacco products and do not address the risks of HTPs. We therefore recommend revising current anti-smoking classes by addressing the dangers of HTPs.

This study was limited by a number of issues. First, the cross-sectional data in this study was limited to factors associated with the use of IQOS; therefore, we were unable to determine whether the study subjects had used cigarettes or e-cigarettes before using IQOS (i.e., potential causality). Thus, we were able to report only on the correlations among IQOS use, cigarette use, and e-cigarette use. In the future, researchers could conduct follow-up studies to identify the issues underlying IQOS use. Second, the fact that the survey was self-administered means that the actual smoking behavior of the subjects could not be verified, which may have compromised estimates pertaining to the prevalence of IQOS use. Note that we deleted inconsistent answers to tobacco-related questions from the questionnaire. Finally, this survey did not ask the participants how they purchased or otherwise obtained HTP products. In the future, researchers could focus on the specific types of HTP being used by young people as well as information pertaining to pricing and availability.

Currently, Taiwan is not importing HTPs. (i.e., HTPs are illegal). According to Article 14 of the Tobacco Hazards Prevention Act, "No person shall manufacture, import, or sell candies, snacks, toys, or any other objects in form of tobacco products." HTPs that contain nicotine are treated as a drug (i.e., a violation of the Pharmaceutical Affairs Act). The Tobacco Hazards Prevention Act was passed in 2009, and since that time, no new regulations pertaining to new tobacco products (e.g., e-cigarettes, HTPs) have been passed. In view of rapid increases in the use of new type tobacco products, amendments to the Tobacco Hazards Prevention Act are urgently required.

The sale of HTPs is not legally sanctioned in Taiwan; however, the sale of IQOS in other countries (e.g., Japan and South Korea) has made it accessible to young people in Taiwan. Governments should decide as soon as possible whether to permit or ban these products; otherwise young people will become habituated to purchasing HTPs illegally. Governments are currently discussing amendments to the "Tobacco Hazards Prevention Act," which has not been amended since 2009. Due to alarming increases in the prevalence of HTP use among youth, we recommend that the government act as soon as possible to amend the "Tobacco Hazards Prevention Act". We also recommend the inclusion of HTPs in school programs addressing the dangers of tobacco.

**Table 4. Logistic regression results for current use of IQOS among Taiwanese youth (2018).**

| Independent Variable | Y = Current use IQOS | | | | | | | |
|---|---|---|---|---|---|---|---|---|
| | Model 1 | | | | Model 2 | | | |
| | aOR | 95%CI | | P | aOR | 95%CI | | P |
| Use of cigarettes in the previous month | | | | | | | | |
| Yes | 4.09 | 2.94 | 5.68 | ‡ | | | | |
| No | 1 | | | | | | | |
| Use of e-cigarettes in the previous month | | | | | | | | |
| Yes | 6.18 | 4.29 | 8.90 | ‡ | | | | |
| No | 1 | | | | | | | |
| Combined use of cigarettes and e-cigarettes in the previous month | | | | | | | | |
| Use of both cigarettes and e-cigarettes | | | | | 19.98 | 14.09 | 28.33 | ‡ |
| Use of e-cigarettes only | | | | | 13.71 | 8.43 | 22.30 | ‡ |
| Use of cigarettes only | | | | | 6.12 | 4.38 | 8.54 | ‡ |
| None | | | | | 1 | | | |
| Gender | | | | | | | | |
| Boy | 1.36 | 1.08 | 1.72 | † | 1.32 | 1.05 | 1.66 | * |
| Girl | 1 | | | | 1 | | | |
| Grade | | | | | | | | |
| 7 | 1 | | | | 1 | | | |
| 8 | 0.81 | 0.55 | 1.19 | | 0.79 | 0.53 | 1.18 | |
| 9 | 0.73 | 0.49 | 1.08 | | 0.73 | 0.49 | 1.08 | |
| 10 | 0.66 | 0.47 | 0.94 | * | 0.65 | 0.45 | 0.93 | * |
| 11 | 0.43 | 0.29 | 0.64 | ‡ | 0.44 | 0.29 | 0.64 | ‡ |
| 12 | 0.52 | 0.35 | 0.77 | † | 0.51 | 0.34 | 0.77 | † |
| Monthly Allowance | | | | | | | | |
| USD 0 | 1.01 | 0.66 | 1.53 | | 1.07 | 0.70 | 1.62 | |
| USD ≦50 | 0.66 | 0.49 | 0.88 | † | 0.68 | 0.51 | 0.91 | * |
| USD 50–100 | 0.83 | 0.62 | 1.12 | | 0.85 | 0.63 | 1.14 | |
| USD ≧100 | 1 | | | | 1 | | | |
| Educational level of parents | | | | | | | | |
| Junior high school and below | 1 | | | | 1 | | | |
| Senior high school | 1.06 | 0.78 | 1.46 | | 1.06 | 0.78 | 1.43 | |
| University or college | 0.90 | 0.63 | 1.29 | | 0.91 | 0.64 | 1.28 | |
| Graduate school | 0.96 | 0.60 | 1.56 | | 0.98 | 0.62 | 1.54 | |
| Smoking status at home | | | | | | | | |
| Both parents smoke | 0.82 | 0.54 | 1.23 | | 0.80 | 0.53 | 1.21 | |
| Only the father smokes | 0.93 | 0.71 | 1.23 | | 0.92 | 0.70 | 1.20 | |
| Only the mother smokes | 1.25 | 0.62 | 2.54 | | 1.15 | 0.57 | 2.34 | |
| Neither smokes | 1 | | | | 1 | | | |
| Smoking status of close friends | | | | | | | | |
| Yes | 2.38 | 1.78 | 3.18 | ‡ | 2.25 | 1.67 | 3.03 | ‡ |
| No | 1 | | | | 1 | | | |
| Access to free cigarettes | | | | | | | | |
| Yes | 2.50 | 1.68 | 3.73 | ‡ | 2.54 | 1.74 | 3.72 | ‡ |
| No | 1 | | | | 1 | | | |
| Exposure to cigarette advertisements | | | | | | | | |
| Yes | 1.09 | 0.84 | 1.42 | | 1.07 | 0.82 | 1.39 | |
| No | 1 | | | | 1 | | | |

(*Continued*)

**Table 4.** (Continued)

| Independent Variable | Y = Current use IQOS | | | | | |
|---|---|---|---|---|---|---|
| | Model 1 | | | Model 2 | | |
| | aOR | 95%CI | | P | aOR | 95%CI | | P |
| Exposure to anti-tobacco courses | | | | | | | | |
| Yes | 1.16 | 0.90 | 1.49 | | 1.16 | 0.91 | 1.49 | |
| No | 1 | | | | 1 | | | |

aOR: adjusted odds ratios.

Model 1: Determining the use of cigarettes OR e-cigarettes.

Model 2: Considering the use of cigarettes AND e-cigarettes.

*P<0.05

†P<0.01

‡P<0.001.

## Acknowledgments

We would like to thank the Health Promotion Administration, Ministry of Health and Welfare, for providing the data used in this study. The content of this research does not represent the opinion(s) of the Health Promotion Administration, Ministry of Health and Welfare.

## Author Contributions

**Conceptualization:** Yue-Chune Lee, Pei-Ching Chen.

**Formal analysis:** Li-Chuan Chang, Chieh Hsu.

**Investigation:** Li-Chuan Chang, Chieh Hsu.

**Methodology:** Pei-Ching Chen.

**Project administration:** Yue-Chune Lee.

**Supervision:** Yue-Chune Lee, Pei-Ching Chen.

**Writing – original draft:** Li-Chuan Chang, Chieh Hsu, Pei-Ching Chen.

**Writing – review & editing:** Yue-Chune Lee, Pei-Ching Chen.

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
