## [Decision Letter · Decision Letter 0]

23 Jul 2020

PONE-D-20-17782

Prevalence of Heated Tobacco Product Use among adolescents in Taiwan

PLOS ONE

Dear Dr. Chen,

Thank you for submitting your manuscript to PLOS ONE. After careful consideration, we feel that it has merit but does not fully meet PLOS ONE’s publication criteria as it currently stands. Therefore, we invite you to submit a revised version of the manuscript that addresses the points raised during the review process.

We look forward to receiving your revised manuscript.

Kind regards,

Stanton A. Glantz

Academic Editor

PLOS ONE

Journal Requirements:

2. Please refer to the specific statistical analyses performed as well as any post-hoc corrections to correct for multiple comparisons. If these were not performed please justify the reasons. Please refer to our statistical reporting guidelines for assistance (https://journals.plos.org/plosone/s/submission-guidelines.#loc-statistical-reporting).

3. We noted in your submission details that a portion of your manuscript may have been presented or published elsewhere.

[A related manuscript entitled “Comparing the characteristics of cigarette smoking and e-cigarette and IQOS use among adolescents in Taiwan” currently in press in Journal of Environmental and Public Health.]

4.We note that you have indicated that data from this study are available upon request. PLOS only allows data to be available upon request if there are legal or ethical restrictions on sharing data publicly. For information on unacceptable data access restrictions, please see http://journals.plos.org/plosone/s/data-availability#loc-unacceptable-data-access-restrictions.

Reviewers' comments:

Reviewer's Responses to Questions

**Comments to the Author**

1. Is the manuscript technically sound, and do the data support the conclusions?

Reviewer #1: Partly

Reviewer #2: Partly

2. Has the statistical analysis been performed appropriately and rigorously? 

Reviewer #1: Yes

Reviewer #2: I Don't Know

3. Have the authors made all data underlying the findings in their manuscript fully available?

Reviewer #1: No

Reviewer #2: No

4. Is the manuscript presented in an intelligible fashion and written in standard English?

Reviewer #1: No

Reviewer #2: Yes

5. Review Comments to the Author

Reviewer #1: I found that the paper, titled “Prevalence of Heated Tobacco Product Use among adolescents in Taiwan”, was interesting to read. Because although HTP products were not sold legally in Taiwan, there were the students who had already experienced in using IQOS and who were currently using it. The findings of this paper can be useful, but I suppose the authors should revise and improve the manuscript based on the comments below.

Major comments

I think the manuscript is too brief and has missed some of the useful information for the readers. I suggest that the authors should explain in detail how the students can buy or get HTP products in the Taiwanese tobacco market where does not allow HTP use and also provide more detailed information about HTP black market in Taiwan.

In the discussion section, the authors compared their findings with the Korean case. The comparison was only focusing on the prevalence of HTPs among the Korean youth and the Taiwanese youth. It is better to discuss the difference between the Korean HTP market and the Taiwanese HTP market.

The authors mentioned several research limitations, but they did not discuss how they had overcome them.

There was no information regarding IRB.

Minor comments

Abstract

Cigarettes can be revised to conventional cigarettes.

Methods

It will be better to add a figure regarding the sampling selection process.

More information about the GYTS study need to be added.

Results

95% CI should be added after OR

Reviewer #2: This paper describes a study aimed at assessing the awareness and use of the IQOS heated tobacco product system by Taiwanese youth. As PMI seeks to expand IQOS to new markets around the world and has recently garnered US FDA authorization to market IQOS in the USA with a reduced exposure statement, the current study is timely as regulators, policy makers, and others seek independent data on the impact of this product on youth tobacco use. The findings of this study, notably 2% were currently using and 4% had used, are quite surprising given that IQOS was not legally marketed in Taiwan. Other studies have found IQOS use by youth but not to my knowledge this much use in a country where it is not legal to market IQOS. This is a noteworthy finding that would likely be of much interest and concern to tobacco control folks. The data source is a large sample of students from the Taiwan (2018) GYTS and is appropriate for the research questions of interest. The primary limitation of this study is that it does not explain how the substantial number of Taiwanese youth IQOS users are obtaining IQOS and for current users, their supply of Heatsticks (HEETS), when the product is not legally marketed there. If there are GYTS data that could illuminate this, I would encourage their inclusion in the paper. In the absence of such data, the authors could perhaps note this as a limitation and offer informed hypotheses for future research in the discussion section. Additional, generally minor, comments and suggestions are provided below:

1. Abstract and introduction (1st paragraph): It has been claimed that IQOS is or was an acronym for “I-Quit_Ordinary-Smoking” but PMI folks have denied this and I’ve seen no evidence. Unless the authors have evidence of this meaning of IQOS that they can cite, I suggest removing it.

2. Introduction, 1st paragraph: This is a narrow definition of heated tobacco products. Some HTPs do not use sticks (PAX, Ploom Tech).

3. Introduction, 2nd paragraph: Please note the nationality and year for these studies. Kim et al. was of a S. Korean population. It is also worth noting for this study that it was an online sample and not designed to yield population estimates. For the Czoli et al. study, please provide year. It may also be worth noting that when this study was conducted (2017), IQOS was not legally marketed in the US (like Taiwan in the present study).

4. Introduction, 3rd paragraph: This section could be much stronger and more relevant. For instances, a link between e-cigarette use and current allergic rhinitis is not directly relevant to the current study nor is the sentence on EVALI. A brief summary of the literature on the toxicity and health effects of IQOS would be fine to highlight the significance of this research, but as the health effects of IQOS are not the subject of this study, the health effects shouldn’t be belabored. Rather, the manuscript space could be used to provide better coverage of the research literature that is more relevant. For instance, there is a study that examined youth use/awareness in Italy and reports documenting the youth appealing marketing of IQOS (for example, see Reuters investigative report on IQOS use of young social media influencers and advertising captured by the Campaign for Tobacco Free Kids).

5. Introduction, final paragraph (p. 6): Please provide a citation for and discuss the study that found that “the prevalence of IQOS use by young people is far from negligible.” If referring to the present study, then please remove the sentence, as it is not common to report the findings/conclusions in the introduction section.

6. p. 8 (“After sample weighting,… using an IQOS product”). First, it is atypical to provide results in the method section. But more critically, these estimates of IQOS use do not make sense as the frequencies are more than one-half the size of the 15-24 year-old Taiwanese population. The sentence is also grammatically awkward (“had answering”) and estimates are not accompanied by 95% confidence intervals.

7. P. 8, Variables (1st paragraph): Was a picture and/or description of IQOS provided? In US e-cigarette research with youth, it has been commonly observed that providing a description with pictures improves measurement of awareness and use. If a description/picture were not provided, how do we know that some youth are not confusing IQOS for something else?

8. P. 9 (“The smoking status among close friends was categorized as… no choice”) – it is not clear what this means.

9. P. 11 (“A comparatively high percentage…”): This sentence is nearly identical to the prior sentence.

10. P. 12 (“Low/moderate allowance was inversely correlated with the likelihood of previously trying IQOS”): It is hard to interpret the direction of this finding and the two odds ratios provided are not clear. In general, for clearer communication of the results, I suggest a structure along the lines of: “Having X was associated with an X times greater (or lower) odds of Y.”

11. Discussion (“The usage statistics… similar report on young Korean adults by Kim et al.”): I would not compare the results of this study with the Kim et al. study. Besides the different age population, the Kim et al. study was of a relatively small, online nonprobability sample. The amount of youth use in this study is surprising given it isn’t legally marketed in Taiwan and greater use in Taiwan than in S. Korea would be similarly surprising because it is legally marketed there. But, estimates are simply not comparable between the two studies.

12. In Table 1, what was striking was that among those who previously tried IQOS, 78% were not past month smokers and 88% were not past month e-cigarette users. So, while smoking is predictive of IQOS use, the majority of prior (and current) IQOS users were not current/recent smokers or e-cigarette users. This warrants closer examination and discussion as it has potential implications for the population health impact of IQOS.

6. PLOS authors have the option to publish the peer review history of their article (what does this mean?). If published, this will include your full peer review and any attached files.

Reviewer #1: No

Reviewer #2: No

---

## [Author Response · Author response to Decision Letter 0]

28 Sep 2020

Review Comments to the Author

Reviewer #1: I found that the paper, titled “Prevalence of Heated Tobacco Product Use among adolescents in Taiwan”, was interesting to read. Because although HTP products were not sold legally in Taiwan, there were the students who had already experienced in using IQOS and who were currently using it. The findings of this paper can be useful, but I suppose the authors should revise and improve the manuscript based on the comments below.

Major comments

I think the manuscript is too brief and has missed some of the useful information for the readers. I suggest that the authors should explain in detail how the students can buy or get HTP products in the Taiwanese tobacco market where does not allow HTP use and also provide more detailed information about HTP black market in Taiwan.

Response: To the Introduction, we have added further details pertaining to the HTP market in Taiwan. Despite the fact that HTP products are not sold legally in Taiwan, many young people still have access to them via the internet (e.g., Facebook groups, LINE groups, etc.). There are also a few large-scale e-cigarette or IQOS stores on the market that operate brick-and mortar stores. 

In the discussion section, the authors compared their findings with the Korean case. The comparison was only focusing on the prevalence of HTPs among the Korean youth and the Taiwanese youth. It is better to discuss the difference between the Korean HTP market and the Taiwanese HTP market.

Response: We have detailed the difference between the Korean and Taiwanese HTP markets in the Discussion section.

Our usage statistics in the current study are slightly lower than those in a report on young Korean adults by Kim et al., in which 5.7% of the study population had previously tried IQOS and 3.5% were current users. The differences can be attributed largely to the study methods (school-based survey vs. online survey), the size of the sample, and the age of the respondents (12–18 years vs. 19–24 years). Nonetheless, we believe that the main reason for the discrepancy is a difference in attitudes toward IQOS use. IQOS products were introduced in Korea in 2017, and the devices are currently available in convenience stores. By contrast, Taiwan forbids the sale of IQOS, with the result that people must resort to online sales or illegal purchases. It should be noted that despite these impediments to accessing the product, the difference in usage behavior between the two countries is really very small, indicating that this issue requires attention.

The authors mentioned several research limitations, but they did not discuss how they had overcome them.

Response: We have rewritten the discussion of limitations in the Discussion section.

This study was limited by a number of issues. First, the cross-sectional data in this study was limited to factors associated with the use of IQOS; therefore, we were unable to determine whether the study subjects had used cigarettes or e-cigarettes before using IQOS (i.e., potential causality). Thus, we were able to report only on the correlations among IQOS use, cigarette use, and e-cigarette use. In the future, researchers could conduct follow-up studies to identify the issues underlying IQOS use. Second, the fact that the survey was self-administered means that the actual smoking behavior of the subjects could not be verified, which may have compromised estimates pertaining to the prevalence of IQOS use. Note that we deleted inconsistent answers to tobacco-related questions from the questionnaire. Finally, this survey did not ask the participants how they purchase or otherwise obtain HTP products. In the future, researchers could focus on the specific types of HTP being used by young people as well as information pertaining to pricing and availability.

There was no information regarding IRB.

Response: This study used secondary data provided by the Health Promotion Administration of Taiwan, and ethical approval for the current study was granted by the Institutional Review Board of National Yang-Ming University (YM108025E). As for the conduct of the Taiwan Global Youth Tobacco Survey, it is an official investigation implemented annually by the Health Promotion Administration, which has also passed the IRB review (EC1041011-F-R2). We have included this information in the Methods section.

Minor comments

Abstract

Cigarettes can be revised to conventional cigarettes.

Methods

It will be better to add a figure regarding the sampling selection process.

More information about the GYTS study need to be added.

Results

95% CI should be added after OR

Response: We have modified the manuscript in accordance with the above comments.

Reviewer#2: This paper describes a study aimed at assessing the awareness and use

of the IQOS heated tobacco product system by Taiwanese youth. As PMI seeks to expand IQOS to new markets around the world and has recently garnered US FDA authorization to market IQOS in the USA with a reduced exposure statement, the current study is timely as regulators, policy makers, and others seek independent data on the impact of this product on youth tobacco use. The findings of this study, notably 2% were currently using and 4% had used, are quite surprising given that IQOS was not legally marketed in Taiwan. Other studies have found IQOS use by youth but not to my knowledge this much use in a country where it is not legal to market IQOS. This is a noteworthy finding that would likely be of much interest and concern to tobacco control folks. The data source is a large sample of students from the Taiwan (2018) GYTS and is appropriate for the research questions of interest. The primary limitation of this study is that it does not explain how the substantial number of Taiwanese youth IQOS users are obtaining IQOS and for current users, their supply of Heatsticks (HEETS), when the product is not legally marketed there. If there are GYTS data that could illuminate this, I would encourage their inclusion in the paper. In the absence of such data, the authors could perhaps note this as a limitation and offer informed hypotheses for future research in the discussion section. Additional, generally minor, comments and suggestions are provided below:

Response: The GYTS does not inquire about how the participants obtain HTPs. Nonetheless, in the Introduction section, we have added a description of how young people usually obtain HTPs. In the Limitations section, we also recommend that researchers focus on the specific types of HTP being used by young people as well as information pertaining to pricing and availability.

1. Abstract and introduction (1st paragraph): It has been claimed that IQOS is or was an acronym for “I-Quit_Ordinary-Smoking” but PMI folks have denied this and I’ve seen no evidence. Unless the authors have evidence of this meaning of IQOS that they can cite, I suggest removing it.

Response: We have removed the words mentioned above.

2. Introduction, 1st paragraph: This is a narrow definition of heated tobacco products. Some HTPs do not use sticks (PAX, Ploom Tech).

Response: We have modified the definition of HTPs.

3. Introduction, 2nd paragraph: Please note the nationality and year for these studies. Kim et al. was of a S. Korean population. It is also worth noting for this study that it was an online sample and not designed to yield population estimates. For the Czoli et al. study, please provide year. It may also be worth noting that when this study was conducted (2017), IQOS was not legally marketed in the US (like Taiwan in the present study).

Response: We have revised the description in the second paragraph of the Introduction section.

4. Introduction, 3rd paragraph: This section could be much stronger and more relevant. For instances, a link between e-cigarette use and current allergic rhinitis is not directly relevant to the current study nor is the sentence on EVALI. A brief summary of the literature on the toxicity and health effects of IQOS would be fine to highlight the significance of this research, but as the health effects of IQOS are not the subject of this study, the health effects shouldn’t be belabored. Rather, the manuscript space could be used to provide better coverage of the research literature that is more relevant. For instance, there is a study that examined youth use/awareness in Italy and reports documenting the youth appealing marketing of IQOS (for example, see Reuters investigative report on IQOS use of young social media influencers and advertising captured by the Campaign for Tobacco Free Kids).

Response: We have removed this paragraph and added new articles pertaining to the use of IQOS and HTPs in other countries.

5. Introduction, final paragraph (p. 6): Please provide a citation for and discuss the study that found that “the prevalence of IQOS use by young people is far from negligible.” If referring to the present study, then please remove the sentence, as it is not common to report the findings/conclusions in the introduction section.

Response: We have revised this paragraph in the Introduction section. 

6. p. 8 (“After sample weighting,… using an IQOS product”). First, it is atypical to provide results in the method section. But more critically, these estimates of IQOS use do not make sense as the frequencies are more than one-half the size of the 15-24 year-old Taiwanese population. The sentence is also grammatically awkward (“had answering”) and estimates are not accompanied by 95% confidence intervals.

Response: The above-mentioned sentences describe the weighted sample size of the two dependent variables. The estimates of IQOS use can be explained by the difference in the number of respondents who provided answers pertaining to the two dependent variables. We have revised the sentences to clarify this issue and added a figure (Fig. 1) illustrating the sample selection process.

After sample weighting, the number of students who responded to the ever use IQOS item (“Have you ever used an IQOS?”) was equivalent to 1,498,502 in the general population. Likewise, the number of students who responded to the current use IQOS item (“During the past 30 days, have you used an IQOS?”) was equivalent to 1,501,527 students in the general population.

7. P. 8, Variables (1st paragraph): Was a picture and/or description of IQOS provided? In US e-cigarette research with youth, it has been commonly observed that providing a description with pictures improves measurement of awareness and use. If a description/picture were not provided, how do we know that some youth are not confusing IQOS for something else?

Response: No images of the IQOS device were included in the research questionnaire; however, IQOS was indicated as a heated tobacco product. Note also that electronic cigarettes and IQOS were discussed using separate questions.

8. P. 9 (“The smoking status among close friends was categorized as… no choice”) – it is not clear what this means.

9. P. 11 (“A comparatively high percentage…”): This sentence is nearly identical to the prior sentence.

10. P. 12 (“Low/moderate allowance was inversely correlated with the likelihood of previously trying IQOS”): It is hard to interpret the direction of this finding and the two odds ratios provided are not clear. In general, for clearer communication of the results, I suggest a structure along the lines of: “Having X was associated with an X times greater (or lower) odds of Y.”

Response: We have revised the manuscript in accordance with the above comments.

11. Discussion (“The usage statistics… similar report on young Korean adults by Kim et al.”): I would not compare the results of this study with the Kim et al. study. Besides the different age population, the Kim et al. study was of a relatively small, online nonprobability sample. The amount of youth use in this study is surprising given it isn’t legally marketed in Taiwan and greater use in Taiwan than in S. Korea would be similarly surprising because it is legally marketed there. But, estimates are simply not comparable between the two studies.

Response: We agree that these two studies cannot be compared. As far as the utilization rate is concerned, we have revised the paragraph in accordance with the opinion of the first reviewer.

12. In Table 1, what was striking was that among those who previously tried IQOS, 78% were not past month smokers and 88% were not past month e-cigarette users. So, while smoking is predictive of IQOS use, the majority of prior (and current) IQOS users were not current/recent smokers or e-cigarette users. This warrants closer examination and discussion as it has potential implications for the population health impact of IQOS.

Response: We have changed Table 1 and Table 2 to have the percentages listed in columns. 

We have determined that the use of conventional cigarettes and the use of e-cigarettes are both important factors pertaining to the use of IQOS. In other words, current users of cigarettes or e-cigarettes were more likely to have previously used or be currently using IQOS. 

 We also established a second model in which the independent variables were combined as a single variable indicating smoking in general, which was then divided into the following four categories: cigarette use only, e-cigarette use only, dual-use, and none. 

We have revised the manuscript in the Methods, Results, and Discussion sections to clarify this issue.

---

## [Decision Letter · Decision Letter 1]

5 Nov 2020

PONE-D-20-17782R1

Prevalence of heated tobacco product use among adolescents in Taiwan

PLOS ONE

Dear Dr. Chen,

Thank you for submitting your manuscript to PLOS ONE. After careful consideration, we feel that it has merit but does not fully meet PLOS ONE’s publication criteria as it currently stands. Therefore, we invite you to submit a revised version of the manuscript that addresses the points raised during the review process.

We look forward to receiving your revised manuscript.

Kind regards,

Stanton A. Glantz

Academic Editor

PLOS ONE

Reviewers' comments:

Reviewer's Responses to Questions

**Comments to the Author**

1. If the authors have adequately addressed your comments raised in a previous round of review and you feel that this manuscript is now acceptable for publication, you may indicate that here to bypass the “Comments to the Author” section, enter your conflict of interest statement in the “Confidential to Editor” section, and submit your "Accept" recommendation.

Reviewer #1: All comments have been addressed

Reviewer #2: (No Response)

2. Is the manuscript technically sound, and do the data support the conclusions?

Reviewer #1: Partly

Reviewer #2: Yes

3. Has the statistical analysis been performed appropriately and rigorously? 

Reviewer #1: Yes

Reviewer #2: Yes

4. Have the authors made all data underlying the findings in their manuscript fully available?

Reviewer #1: Yes

Reviewer #2: No

5. Is the manuscript presented in an intelligible fashion and written in standard English?

Reviewer #1: No

Reviewer #2: Yes

6. Review Comments to the Author

Reviewer #1: I have several minor questions and comments. I hope that the authors add their answers to the manuscript.

Introduction

- HTPs usually stands for heated tobacco products. 'Heat-not-burn tobacco product' is the term that the tobacco industry wanted to call it.

- Why does HTP use ban in Taiwan?

- Please provide the year when and where PMI launched IQOS first for the journal readers.

- How much is the IQOS device on the internet? Can the Taiwanese kids afford it? How was it marketed on the internet? Do you have any samples of it?

- Is there only IQOS available? What about glo from BAT, lil from KT&G or many other devices manufactured by China?

- If there are other products apart from IQOS, do you have any reason why you asked only IQOS use?

Discussion

- Price difference can be considered when you compared to Korea and Twain.

Conclusion

- The conclusion of this study is recommending to revise the Taiwanese Tobacco Act. Do you have any insight to provide international readers based on your experience in Taiwan?

Reviewer #2: The authors were responsive to my comments from the initial review and the manuscript is improved. Remaining issues are relatively minor but if addressed would improve this manuscript, mostly for clarity but also in substance.

1. In response to my initial comments, the introduction was revised to remove less relevant material. This is an improvement. However, while containing some additional helpful elaboration of the research, it does not address the existing research on the youth oriented marketing of IQOS. It might also be worth noting the countries in Asia where IQOS is legally sold. Could these be the sources for IQOS black markets in Taiwan? If not, where?

2. Figure 1 was not found in the manuscript.

3. The results, including Tables, are clearer with respect to the direction of effects. However, some text remains less clear, particularly without reference to the Table. Specifically, "The following factors were negatively correlated with the likelihood of previously trying IQOS: low (USD <= 50) and moderate (USD 50-100)...". This is confusing in two aspects. First, this could be misread as higher allowance being associated with lower odds of previously trying IQOS. The authors specify "low (USD <=50) but this is not a standard or clear way of reporting such results. Perhaps more of an issue is that this implies a linear relationship with the relationship as reported in the Table seems to be nonlinear: those with the lowest allowance ($0) were no less likely to use IQOS than those with the highest allowance. Also confusing is that the label for the "low" allowance is USD < 50 even while their is a lower a group with $0. In other words, less than USD50 is not exclusive to USD0.

4. Discussion - "Despite the fact that Taiwan does not allow the sale of IQOS, the usage rates were higher than in the US in 2019, the year in which year the US Food and Drug Administration authorized the marketing of IQOS HTPs.": The article referenced is by Dai et al, which used 2019 NYTS data to characterize youth awareness and use of HTP. Dai et al. did not examine IQOS use. And while IQOS was introduced to the US in 2019, it was late September and only in one market (and only two markets through the end of the year). The 2019 NYTS data were collected in early 2019, prior to the introduction of IQOS to the US market. I would suggest not contrasting the findings of the present study with Dai et al.

5. Lastly - while there are few grammatical or typographical errors, the phrasing of several sentences reads a bit awkward or nonstandard in places. For instance, the sentence ending the first paragraph in the Discussion "use of both cigarettes and e-cigarettes > use of e-cigarette only > use of cigarettes only > none" is a nonstandard (for a Discussion section) use of equality signs (typically would be spelled out). The titles for Tables 3 and 4 are also not standard. For Table 3, for example, consider something more standard such as: Logistic Regression Models for Prior Use of IQOS among Taiwanese Youth (2018). The authors need to use this title; rather, I offer it as an example of a more standard and arguably more informative title. Tables should stand be interpretable separate from the text. These and similar writing issues are more stylistic and minor but if addressed would lead to a more polished paper.

7. PLOS authors have the option to publish the peer review history of their article (what does this mean?). If published, this will include your full peer review and any attached files.

Reviewer #1: No

Reviewer #2: No

---

## [Author Response · Author response to Decision Letter 1]

3 Dec 2020

Review Comments to the Author

Reviewer #1: I have several minor questions and comments. I hope that the authors add their answers to the manuscript.

Introduction

- HTPs usually stands for heated tobacco products. 'Heat-not-burn tobacco product' is the term that the tobacco industry wanted to call it.

Response: We have revised the HTP term to heated tobacco products in the manuscript. 

- Why does HTP use ban in Taiwan?

Response: Currently, Taiwan is not importing HTPs. (i.e., HTPs are illegal). According to Article 14 of the Tobacco Hazards Prevention Act, “No person shall manufacture, import, or sell candies, snacks, toys, or any other objects in form of tobacco products.” HTPs that contain nicotine are treated as a drug (i.e., a violation of the Pharmaceutical Affairs Act). The Tobacco Hazards Prevention Act was passed in 2009, and since that time, no new regulations pertaining to new tobacco products (e.g., e-cigarettes, HTPs) have been passed. In view of rapid increases in the use of new type tobacco products, amendments to the Tobacco Hazards Prevention Act are urgently required.

- Please provide the year when and where PMI launched IQOS first for the journal readers.

Response: Philip Morris International (PMI) introduced the IQOS heated tobacco product in Japan, in November 2014. [1] 

- How much is the IQOS device on the internet? Can the Taiwanese kids afford it? How was it marketed on the internet? Do you have any samples of it?

Response: We searched Taiwanese websites for IQOS products. Generally, IQOS devices are sold on the Internet for 2500-5500 NTD (about 86.8-191.0 USD, where 1 USD = 28.8 NTD), whereas the Cigarette Bomb costs 1000-2100 NTD (34.7-72.9 USD). We determined in this study that only 16% of adolescents received more than 100 USD in pocket money per month, which means that most of them would be unable to afford an IQOS device. Those that can afford the devices can browse pictures and descriptions of various HTP products and make purchases online. After placing an IQOS product in the shopping cart, the purchaser contacts the seller by joining a LINE group in order to arrange shipment. We do not have a sample of the IQOS device. The above description has not been included in the article because we did not want to provide detailed instructions as to the purchasing process.

- Is there only IQOS available? What about glo from BAT, lil from KT&G or many other devices manufactured by China?

Response: A survey of relevant Taiwanese websites revealed that IQOS devices were widely sold IQOS, whereas GLO and LIL were more difficult to find.

- If there are other products apart from IQOS, do you have any reason why you asked only IQOS use?

Response: The fact that IQOS is widely sold online implies that it is accessible to many adolescents. Furthermore, the survey was limited to questions pertaining to IQOS use.

Discussion

- Price difference can be considered when you compared to Korea and Taiwan.

Response: We have addressed this point in the Discussion section. It is certainly possible that the price difference may account for some of the differences in adoption. South Korea has legalized the use of IQOS; therefore, the price is lower. These products are still illegal in Taiwan; therefore, the price is higher. 

Conclusion

- The conclusion of this study is recommending to revise the Taiwanese Tobacco Act. Do you have any insight to provide international readers based on your experience in Taiwan?

Response: The sale of HTPs is not legally sanctioned in Taiwan; however, the sale of IQOS from other countries has made it accessible to young people in Taiwan. We believe that governments should decide as soon as possible whether to permit or ban these devices; otherwise, young people will become habituated to purchasing HTPs illegally.

Reviewer #2: The authors were responsive to my comments from the initial review and the manuscript is improved. Remaining issues are relatively minor but if addressed would improve this manuscript, mostly for clarity but also in substance.

1. In response to my initial comments, the introduction was revised to remove less relevant material. This is an improvement. However, while containing some additional helpful elaboration of the research, it does not address the existing research on the youth oriented marketing of IQOS. It might also be worth noting the countries in Asia where IQOS is legally sold. Could these be the sources for IQOS black markets in Taiwan? If not, where?

Response: IQOS is legally sold in Asian countries such as Japan and South Korea [2]. At present, those countries are the primary source of black market IQOS products in Taiwan. Most IQOS sold online are listed as a Japanese product. 

2. Figure 1 was not found in the manuscript.

Response: In accordance with formatting requirements, Figure 1 has been removed from the revised manuscript; however, it is available as a supplementary file. 

3. The results, including Tables, are clearer with respect to the direction of effects. However, some text remains less clear, particularly without reference to the Table. Specifically, "The following factors were negatively correlated with the likelihood of previously trying IQOS: low (USD <= 50) and moderate (USD 50-100)...". This is confusing in two aspects. First, this could be misread as higher allowance being associated with lower odds of previously trying IQOS. The authors specify "low (USD <=50) but this is not a standard or clear way of reporting such results. Perhaps more of an issue is that this implies a linear relationship with the relationship as reported in the Table seems to be nonlinear: those with the lowest allowance ($0) were no less likely to use IQOS than those with the highest allowance. Also confusing is that the label for the "low" allowance is USD < 50 even while their is a lower a group with $0. In other words, less than USD50 is not exclusive to USD0.

Response: We have revised the results to clarify this point. We observed a nonlinear relationship between monthly allowance and previous use of IQOS. Students with low (USD≦50) or moderate (USD 50-100) monthly allowance were less likely to have tried IQOS than were students with high (USD≧100) monthly allowance. We observed no difference between students with no monthly allowance (USD 0) and those with a high (USD≧100) monthly allowance in terms of previously trying IQOS. 

4. Discussion - "Despite the fact that Taiwan does not allow the sale of IQOS, the usage rates were higher than in the US in 2019, the year in which year the US Food and Drug Administration authorized the marketing of IQOS HTPs.": The article referenced is by Dai et al, which used 2019 NYTS data to characterize youth awareness and use of HTP. Dai et al. did not examine IQOS use. And while IQOS was introduced to the US in 2019, it was late September and only in one market (and only two markets through the end of the year). The 2019 NYTS data were collected in early 2019, prior to the introduction of IQOS to the US market. I would suggest not contrasting the findings of the present study with Dai et al.

Response: Thank you for reminding us about the time at which 2019 NYTS data was collected and the time that IQOS was authorized for sale in the US. Dai [3] reported that IQOS was only authorized for sale in the US during the survey period (February 15, 2019 – May 24, 2019) and that IQOS was not launched in the US until October 2019. Furthermore, Dai examined the ever use and current use of HTPs, including iQOS, glo, and Eclipse. We have updated the discussion in the revised manuscript. 

The usage statistics in the current study are slightly higher than those in Dai’s 2019 report on US students, in which 2.4% of the study population had tried HTPs (including iQOS, glo, and Eclipse) and 1.6% were currently using HTPs. At the time of that study, the sale of IQOS was not sanctioned in Taiwan or the US; however, the usage rates were higher in Taiwan than in the US. Clearly, the use of IQOS products by young people in Taiwan is far from negligible.

5. Lastly - while there are few grammatical or typographical errors, the phrasing of several sentences reads a bit awkward or nonstandard in places. For instance, the sentence ending the first paragraph in the Discussion "use of both cigarettes and e-cigarettes > use of e-cigarette only > use of cigarettes only > none" is a nonstandard (for a Discussion section) use of equality signs (typically would be spelled out). The titles for Tables 3 and 4 are also not standard. For Table 3, for example, consider something more standard such as: Logistic Regression Models for Prior Use of IQOS among Taiwanese Youth (2018). The authors need to use this title; rather, I offer it as an example of a more standard and arguably more informative title. Tables should stand be interpretable separate from the text. These and similar writing issues are more stylistic and minor but if addressed would lead to a more polished paper.

Response: We have revised the Discussion section and Tables in the manuscript.

References

1. Tabuchi T, Kiyohara K, Hoshino T, Bekki K, Inaba Y, Kunugita N. Awareness and use of electronic cigarettes and heat-not-burn tobacco products in Japan. Addiction. 2016;111(4):706-13. Epub 2015/11/15. doi: 10.1111/add.13231. PubMed PMID: 26566956.

2. Glantz SA. Heated tobacco products: the example of IQOS. Tob Control. 2018;27(Suppl 1):s1-s6. Epub 2018/10/26. doi: 10.1136/tobaccocontrol-2018-054601. PubMed PMID: 30352841; PubMed Central PMCID: PMCPMC6252052.

3. Dai H. Heated tobacco product use and associated factors among U.S. youth, 2019. Drug Alcohol Depend. 2020;214:108150. Epub 2020/07/10. doi: 10.1016/j.drugalcdep.2020.108150. PubMed PMID: 32645682.

---

## [Editor Report · Decision Letter 2]

7 Dec 2020

Prevalence of heated tobacco product use among adolescents in Taiwan

PONE-D-20-17782R2

Dear Dr. Chen,

We’re pleased to inform you that your manuscript has been judged scientifically suitable for publication and will be formally accepted for publication once it meets all outstanding technical requirements.

Kind regards,

Stanton A. Glantz

Academic Editor

PLOS ONE
---

## [Editor Report · Acceptance letter]

10 Dec 2020

PONE-D-20-17782R2 

Prevalence of heated tobacco product use among adolescents in Taiwan 

Dear Dr. Chen:

I'm pleased to inform you that your manuscript has been deemed suitable for publication in PLOS ONE. Congratulations! Your manuscript is now with our production department. 

Kind regards, 

on behalf of

Professor Stanton A. Glantz 

Academic Editor

PLOS ONE